

# The highly rearranged mitochondrial genomes of three economically important scale insects and the mitochondrial phylogeny of Coccoidea (Hemiptera: Sternorrhyncha)

Hong-Ling Liu[1], Qing-Dong Chen[1], Song Chen[1], De-Qiang Pu[1], Zhi-Teng Chen[2], Yue-Yue Liu[3] and Xu Liu[1]

[1] Institute of plant protection, Sichuan Academy of Agricultural Sciences, Key Laboratory of integrated pest management of Southwest crops, Ministry of Agriculture, Chengdu, China
[2] School of Grain Science and Technology, Jiangsu University of Science and Technology, Zhenjiang, China
[3] Analysis and testing center of Sichuan Academy of Agricultural Sciences, Chengdu, China

Corresponding author
Xu Liu, liuxu6186@126.com

## ABSTRACT

The mitochondrial genomes (mitogenomes) of scale insects are less known in comparison to other insects, which hinders the phylogenetic and evolutionary studies of Coccoidea and higher taxa. Herein, the complete mitogenomes of *Unaspis yanonensis*, *Planococcus citri* and *Ceroplastes rubens* were sequenced for Coccoidea. The 15,220-bp long mitogenome of *U. yanonensis* contained the typical set of 37 genes including 13 PCGs, 22 tRNA genes and two rRNA genes; the 15,549-bp long mitogenome of *P. citri* lacked the tRNA gene *trnV*; the 15,387-bp long mitogenome of *C. rubens* exhibited several shortened PCGs and lacked five tRNA genes. The mitochondrial gene arrangement of the three mitogenomes was different from other scale insects and *Drosophila yakuba*. Most PCGs used standard ATN (ATA, ATT, ATC and ATG) start codons and complete TAN (TAA or TAG) termination codons. The *ND4L* had the highest evolutionary rate but *COX1* and *CYTB* were the lowest. Most tRNA genes had cloverleaf secondary structures, whereas the reduction of dihydrouridine (DHU) arms and TψC arms were detected. Tandem repeats, stem-loop (SL) structures and poly-[TA]n stretch were found in the control regions (CRs) of the three mitogenomes. The phylogenetic analyses using Bayesian inference (BI) and maximum likelihood methods (ML) showed identical results, both supporting the inner relationship of Coccoidea as Coccidae + (Pseudococcidae + Diaspididae).

## INTRODUCTION

The scale insects (Coccoidea) are well-known sap-sucking hemipterans which are economically important pests causing severe damage to native crops and plants (*Kondo, Gullan & Williams, 2008*). Adult males of Coccoidea are hyperpaurometamorphosis, whereas the adult females are paurometamorphosis and resemble their nymphs

(*Gullan & Kosztarab, 1997*). These insects are usually smaller than 5 mm and often appear similar color with their host plants. Most scale insects can produce waxy secretion covering their bodies as a protection armature (*Gullan & Kosztarab, 1997*), which also causes difficulty in using chemical control methods.

When compared with other superfamilies of the monophyletic suborder Sternorrhyncha: Aphidoidea (aphids), Aleyrodoidea (whiteflies) and Psylloidea (jumping plant lice), the superfamily Coccoidea possess a higher biodiversity and morphological variety (*Gullan & Martin, 2003*; *Gullan & Cook, 2007*). Despite the previous morphological and molecular contributions (*Koteja, 1974*; *Von Dohlen & Moran, 1995*; *Gullan & Cook, 2007*; *Cook, Gullan & Trueman, 2002*; *Hodgson & Hardy, 2013*), the scale insect systematics especially the family-level classification still remains unresolved.

Morphology of scale insects has apparent limits when used for resolving the higher-level phylogeny of scale insects, which is expected to be improved by the DNA sequence data. Mitochondrial genome (mitogenome) usually contains a typical set of 37 genes: 13 protein-coding genes (PCG), 22 transfer RNA genes (tRNA), two ribosomal RNA genes (rRNA) and a non-coding control region (CR) and has become one of the most popular molecules used in insect phylogenetic studies (*Cameron, 2014*). Recently, *Deng, Lu & Huang (2019)* and *Lu, Huang & Deng (2020)* respectively sequenced the mitogenomes of the two scale insects, *Ceroplastes japonicus* (*Green, 1921*) and *Saissetia coffeae* (*Walker, 1852*) and investigated the efficiency of using mitogenome data in the phylogeny of Sternorrhyncha. Mitochondrial gene rearrangement and truncation of tRNA genes have been found in the two mitogenomes. To facilitate the resolution of phylogeny and molecular evolution of Coccoidea, we sequenced the complete mitogenomes of *Unaspis yanonensis* (*Kuwana, 1923*), *Planococcus citri* (*Risso, 1813*) and *Ceroplastes rubens* (*Maskell, 1893*), which includes the first representatives of Pseudococcidae and Diaspididae. The mitogenomic organizations, gene rearrangements, nucleotide compositions, codon usages of PCGs, secondary structures of tRNA genes and CR were analyzed for the three mitogenomes. In addition, the phylogenetic relationships of four species of Coccoidea were reconstructed to evaluate the validity of the newly obtained molecular data.

## MATERIALS & METHODS

### Sample preparation and DNA extraction

The specimens of *U. yanonensis*, *P. citri* and *C. rubens* were collected from Chengdu, Sichuan Province of China in October of 2019. The specimens were reliably identified by experts of Sichuan Academy of Agricultural Sciences, and were preserved in 100% ethanol. The total genomic DNA of the three scale insects was isolated using the E.Z.N.A.® Tissue DNA Kit (OMEGA, America) and preserved at −20 °C before the sequencing process.

### Sequencing, assembly and annotation

The Illumina TruSeq short-insert libraries (insert size = 450 bp) were constructed using 1.0 μg of purified DNA fragments and sequenced by Illumina Hiseq 4000 (Shanghai BIOZERON Co., Ltd). Prior to assembly, raw reads were filtered and high-quality reads were retained and assembled into contigs by SOAPdenovo2.04 (*Luo et al., 2012*). Then the

assembled contigs were aligned to the reference mitogenome of *C. japonicus* (GenBank accession number MK847519) using BLAST. The aligned contigs (≥80% similarity and query coverage) were arranged according to the reference mitogenome. Finally, the clean reads were mapped to the assembled draft mitogenome to fix the wrong bases; gaps were filled using GapFiller v2.1.1 (https://sourceforge.net/projects/gapfiller/). The mitogenome sequences of *U. yanonensis*, *P. citri* and *C. rubens* were deposited in GenBank under the accession numbers MT611525, MT611526 and MT677923, respectively.

Most tRNA genes were predicted and depicted by MITOS (*Bernt et al., 2013*); structures of several tRNA genes of *C. rubens* were predicted manually. PCGs and rRNA genes were identified by homology alignments. Gene boundaries of PCGs were confirmed in ORF finder (https://www.ncbi.nlm.nih.gov/orffinder/). The graphic view of the mitogenomes were computed using CGView Server (http://stothard.afns.ualberta.ca/cgview_server/) (*Grant & Stothard, 2008*). The probable mitochondrial rearrangement scenarios during the evolution of *U. yanonensis*, *P. citri* and *C. rubens* were predicted by the CREx (Common Interval Rearrangement Explorer) online server (*Bernt, 2007*) using *Drosophila yakuba* as a reference (*Clary & Wolstenholme, 1985*). Nucleotide composition of each gene and codon usage of PCGs were calculated by MEGA v.6.0 (*Tamura et al., 2013*). The composition skew analysis was conducted by AT-skew = $[A-T]/[A+T]$ and GC-skew = $[G-C]/[G+C]$ formulas (*Perna & Kocher, 1995*). The software DnaSP v. 5.10 (*Librado & Rozas, 2009*) was used to calculate the synonymous substitution rate (Ks) and the nonsynonymous substitution rate (Ka). Presumed secondary structures in the control region were predicted by the online tool Tandem Repeats Finder (http://tandem.bu.edu/trf/trf.advanced.submit.html) and DNAMAN v6.0.3.

## Phylogenetic analysis

Nucleotide sequences of PCGs derived from four species of Coccoidea, including *U. yanonensis*, *P. citri* and *C. rubens* sequenced in this study, were used in the phylogenetic analysis (Table 1). The species *S. coffeae* was not included in the dataset due to the unannotated and unreliable status of its sequence as noted in Genbank. The two aphids, *Aphis glycines* and *Diuraphis noxia* were used as the outgroups. The 13 PCGs were aligned by MAFFT and concatenated as a combined dataset using SequenceMatrix v1.7.8 (*Katoh & Standley, 2013*). PartitionFinder v2.1.1 was used to determine the optimal nucleotide substitution models and partitioning schemes by using the Bayesian Information Criterion (BIC) and a greedy search algorithm (*Lanfear et al., 2016*). Two phylogenetic inferences were conducted with the partition schemes, including Bayesian inferences (BI) and Maximum likelihood (ML) analysis. BI analysis was conducted by MrBayes v3.2.7, with 10 million generations sampling every 1,000 generations, running one cold chain and three hot chains with a burn-in of 25% trees (*Ronquist & Huelsenbeck, 2003*). Stability of the results of BI analysis was examined by Tracer v.1.5. ML analysis was performed by RAxML v8.2.12 with 1,000 bootstrap replicates (*Stamatakis, 2014*). Tree files generated by both BI and ML trees were adjusted and visualized in FigTree v1.4.2.
**Table 1  Species of Hemiptera used in this study.**

| Superfamily | Family | Species | Accession number |
|---|---|---|---|
| Coccoidea | Coccidae | *Ceroplastes japonicus* | MK847519 |
| | | *Ceroplastes rubens* | MT677923 |
| | Diaspididae | *Unaspis yanonensis* | MT611525 |
| | Pseudococcidae | *Planococcus citri* | MT611526 |
| Aphidoidea | Aphididae | *Aphis glycines* | MK111111 |
| | | *Diuraphis noxia* | KF636758 |

## RESULTS

### Mitogenome annotation and nucleotide composition

The complete mitogenomes of *U. yanonensis*, *P. citri* and *C. rubens* were all typical double-strand circular molecules with a length of 15,220 bp, 15,549 bp and 15,387 bp, respectively (Fig. 1), which were similar to other mitogenomes of Coccoidea (*Deng, Lu & Huang, 2019*; *Lu, Huang & Deng, 2020*). The standard set of 37 genes (13 PCGs, 22 tRNA genes and two rRNA genes) were all found in the mitogenome of *U. yanonensis* (Table 2), whereas *trnV* was lost in *P. citri* (Table 3); *C. rubens* lacked five tRNA genes, *trnC*, *trnR*, *trnS2*, *trnL1* and *trnV* (Table 4). In *U. yanonensis*, there were nine overlapping nucleotides located in four pairs of neighboring genes (Table 2); while in *P. citri*, there were 36 overlapping nucleotides in nine gene boundaries (Table 3). In *C. rubens*, there were only seven overlapping nucleotides in four gene boundaries (Table 4). The longest overlap was 18-bp long and located between *trnS2* and *ND1* in *P. citri*. There were 227 intergenic nucleotides (IGNs) dispersed in 20 locations for *U. yanonensis*, 126 IGNs in 19 locations for *P. citri* and 478 IGNs in 19 locations for *C. rubens*, indicating a loose structure of the three scale insect mitogenomes.

The whole mitogenomes of *U. yanonensis*, *P. citri* and *C. rubens* were strongly biased toward A and T nucleotides (86.6%, 82.7% and 87.5%, respectively). The *U. yanonensis* mitogenome had negative AT-skew and positive GC-skew, whereas *P. citri* and *C. rubens* exhibited positive AT-skew and negative GC-skew. The A+T contents were also rich in the mitochondrial genes, showing the highest in *trnF* of *U. yanonensis* and *P. citri*, and *trnG* of *C. rubens*.

### Gene rearrangement

The mitochondrial genes of *U. yanonensis*, *P. citri* and *C. rubens* were highly rearranged, being different from the two sequenced scale insects, *C. japonicus* and *S. coffeae* (*Deng, Lu & Huang, 2019*; *Lu, Huang & Deng, 2020*). When compared with *D. yakuba*, *U. yanonensis* and *P. citri* both showed a conserved gene cluster *trnE-trnF- ND5-trnH- ND4- ND4l-trnT-trnP- ND6-CYTB-trnS2- ND1-trnL1-rrnL*; *C. rubens* had three shorter conserved gene clusters, *COX1-trnL-COX2-trnK-trnD*, *COX3-trnG-ND3* and *ND5-trnH-ND4-ND4L* (Fig. 2). The mitogenome of *U. yanonensis* exhibited the rearrangement of three cytochrome c oxidase subunit genes (*COX1*, *COX2*, *COX3*), two NADH dehydrogenase subunit genes (*ND2* and *ND3*) and many tRNA genes. Despite the multiple tRNA gene rearrangements, the mitogenome of *P. citri* also had a reversal of the ancestral gene cluster *COX1- COX2-ATP8-ATP6- COX3- ND3*. The mitogenome of *C. rubens* showed fewer rearrangements

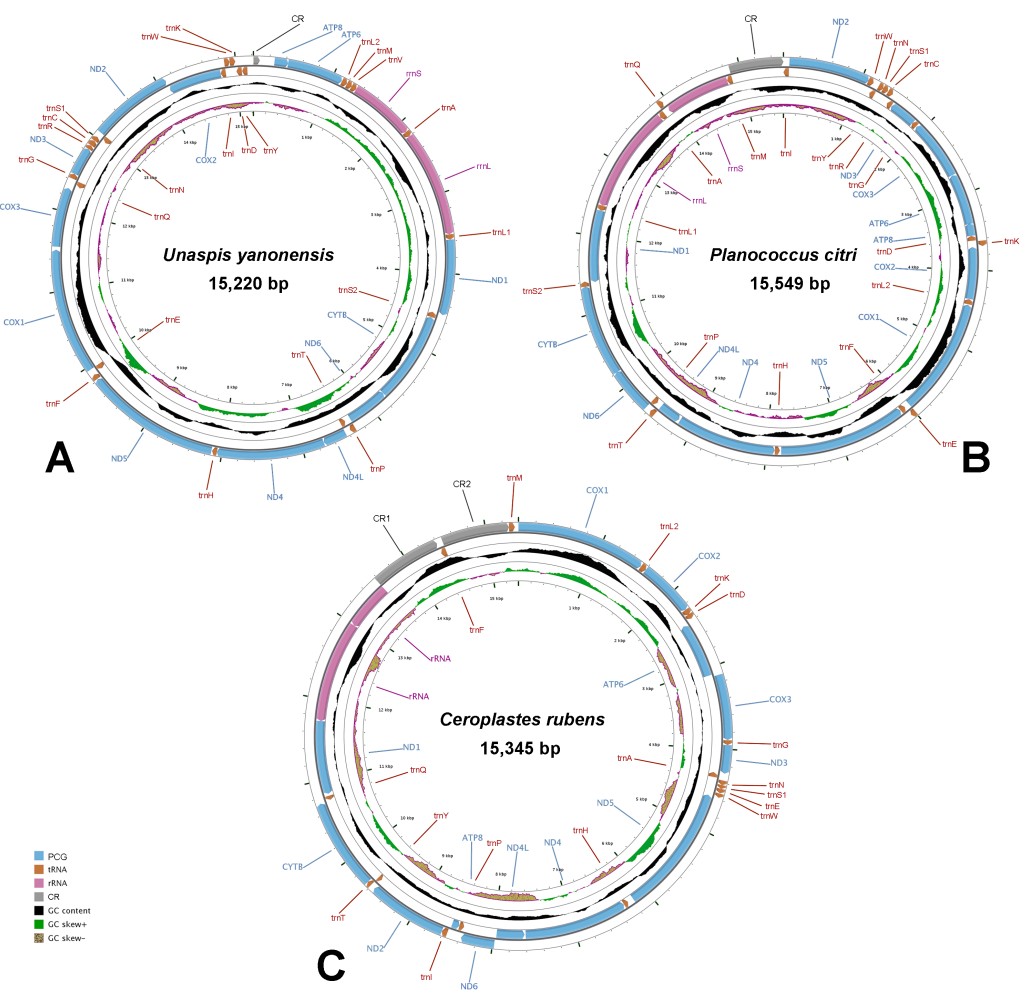

**Figure 1** **Mitochondrial maps of *Unaspis yanonensis*, *Planococcus citri* and *Ceroplastes rubens*.** (A) *Unaspis yanonensis*; (B) *Planococcus citri*; (C) *Ceroplastes rubens*. Genes outside the map are transcribed clockwise, whereas those inside the map are transcribed counterclockwise. The inside circles show the GC content and the GC skew. GC content and GC skew are plotted as the deviation from the average value of the entire sequence.

than *U. yanonensis* and *P. citri*, including two PCGs (*ND2* and *ATP8*) and multiple tRNA genes.

The CREx analysis predicted the alternative scenarios how the three scale insect mitogenomes rearranged from the ancestral type of mitogenome of *D. yakuba* (Figs. 3–5). The mitochondrial gene order of *U. yanonensis* changed from *D. yakuba* by nine steps of rearrangement events, including the transposition of *trnV* and *rrnS*, the subsequent reverse transposition of *trnK* and *trnD*, the reversal of *trnS1*, and additional three reversal events and three tandem duplication and random loss (TDRL) events (Fig. 3). In *P. citri*, the first step is the reversal of *trnK*, followed by two alternative scenarios: the first one contained two reversal events, one TDRL event and one transposition event; the second one included three reversal events, two TDRL events and one transposition event (Fig. 4). Fewer rearrangement

**Table 2  Mitochondrial genome structure of *Unaspis yanonensis*.**

| Gene | Position (bp) | Size (bp) | Direction | Intergenic nucleotides | Anti- or start/ stop codons | A+T% |
|---|---|---|---|---|---|---|
| Control region | 1–78 | 260 | + | 78 | – | 81.9 |
| *ATP8* | 261–428 | 168 | + | 0 | ATT/TAA | 93.5 |
| *ATP6* | 429–1121 | 693 | + | 0 | ATT/TAA | 86.0 |
| *trnL2 (UUR)* | 1127–1193 | 67 | + | 5 | TAA | 85.1 |
| *trnMet (M)* | 1196–1260 | 65 | + | 2 | CAT | 89.2 |
| *trnVal (V)* | 1261–1326 | 66 | + | 0 | TAC | 90.9 |
| *rrnS* | 1327–2133 | 807 | + | 0 | – | 90.6 |
| *trnAla (A)* | 2134–2202 | 69 | + | 0 | TGC | 72.5 |
| *rrnL* | 2203–3516 | 1314 | + | 0 | – | 89.6 |
| *trnLeu1 (CUN)* | 3517–3581 | 65 | + | 0 | TAG | 86.2 |
| *nad1* | 3582–4517 | 936 | + | 0 | ATT/TAA | 84.5 |
| *trnSer2 (UCN)* | 4516–4584 | 69 | – | −2 | TGA | 88.4 |
| *CYTB* | 4587–5759 | 1173 | – | 2 | ATA/TAA | 81.8 |
| *ND6* | 5760–6300 | 541 | – | 0 | ATG/T– | 93.0 |
| *trnPro (P)* | 6306–6373 | 68 | + | 5 | TGG | 91.2 |
| *trnThr (T)* | 6374–6437 | 64 | – | 0 | TGT | 95.3 |
| *ND4L* | 6441–6728 | 288 | + | 3 | ATT/TAA | 89.9 |
| *ND4* | 6731–8062 | 1332 | + | 2 | ATT/TAA | 87.8 |
| *trnHis (H)* | 8062–8121 | 60 | + | −1 | GTG | 93.3 |
| *ND5* | 8131–9810 | 1680 | + | 9 | ATA/TAA | 88.2 |
| *trnPhe (F)* | 9818–9882 | 65 | + | 7 | GAA | 95.4 |
| *trnGlu (E)* | 9890–9957 | 68 | – | 7 | TTC | 95.6 |
| *COX1* | 9959–11515 | 1557 | + | 1 | TTG/TAA | 78.4 |
| *COX3* | 11554–12288 | 735 | + | 38 | ATT/TAA | 82.3 |
| *trnGln (Q)* | 12339–12407 | 69 | – | 50 | TTG | 89.9 |
| *trnGly (G)* | 12417–12479 | 63 | + | 9 | TCC | 88.9 |
| *ND3* | 12480–12830 | 351 | + | 0 | ATT/TAA | 88.3 |
| *trnArg (R)* | 12831–12883 | 53 | + | 0 | TCG | 86.8 |
| *trnCys (C)* | 12885–12954 | 70 | + | 1 | GCA | 94.3 |
| *trnSer1 (AGN)* | 12956–13014 | 59 | + | 1 | GCT | 89.8 |
| *trnAsn (N)* | 13013–13080 | 68 | – | −2 | GTT | 83.8 |
| *ND2* | 13082–14107 | 1026 | + | 1 | ATT/TAA | 92.1 |
| *COX2* | 14104–14793 | 690 | – | −4 | ATT/TAA | 83.6 |
| *trnIle (I)* | 14795–14861 | 67 | – | 1 | GAT | 83.6 |
| *trnTrp (W)* | 14862–14928 | 67 | + | 0 | TCA | 94.0 |
| *trnLys (K)* | 14929–14997 | 69 | + | 0 | CTT | 91.3 |
| *trnAsp (D)* | 15001–15069 | 69 | – | 3 | GTC | 92.8 |
| *trnTyr (Y)* | 15074–15142 | 69 | – | 4 | GTA | 85.5 |

events were predicted in *C. rubens*, including the first step of transposition, the subsequent three reversal events, and final three TDRL events (Fig. 5). Considering the similarly rearranged mitochondrial genes of *C. japonicus* and *S. coffeae*, extensive mitochondrial

**Table 3  Mitochondrial genome structure of *Planococcus citri*.**

| Gene | Position (bp) | Size (bp) | Direction | Intergenic nucleotides | Anti- or start/ stop codons | A+T% |
|---|---|---|---|---|---|---|
| *trnIle (I)* | 1–70 | 70 | − | 0 | GAT | 84.3 |
| *ND2* | 76–1089 | 1014 | + | 5 | ATT/TAA | 87.4 |
| *trnTrp (W)* | 1088–1156 | 69 | + | −2 | TCA | 89.9 |
| *trnTyr (Y)* | 1167–1232 | 66 | − | 10 | GTA | 84.8 |
| *trnAsn (N)* | 1232–1295 | 64 | + | −1 | GTT | 84.4 |
| *trnSer1 (AGN)* | 1295–1359 | 65 | + | −1 | GCT | 80.0 |
| *trnCys (C)* | 1368–1432 | 65 | + | 8 | GCA | 92.3 |
| *trnArg (R)* | 1434–1497 | 64 | − | 1 | TCG | 79.7 |
| *ND3* | 1504–1854 | 351 | − | 6 | ATT/TAA | 84.3 |
| *trnGly (G)* | 1855–1918 | 64 | − | 0 | TCC | 92.2 |
| *COX3* | 1928–2716 | 789 | − | 9 | ATG/TAA | 76.6 |
| *ATP6* | 2721–3395 | 675 | − | 4 | ATG/TAA | 80.1 |
| *ATP8* | 3389–3550 | 162 | − | −7 | ATT/TAA | 85.8 |
| *trnAsp (D)* | 3551–3616 | 66 | − | 0 | GTC | 90.9 |
| *trnLys (K)* | 3629–3695 | 67 | + | 12 | CTT | 86.6 |
| *COX2* | 3700–4380 | 681 | − | 4 | ATT/TAA | 78.6 |
| *trnLeu2 (UUR)* | 4384–4451 | 68 | − | 3 | TAA | 85.3 |
| *COX1* | 4460–5989 | 1530 | − | 8 | ATA/TAA | 74.4 |
| *trnGlu (E)* | 5991–6057 | 67 | + | 1 | TTC | 94.0 |
| *trnPhe (F)* | 6057–6124 | 68 | − | −1 | GAA | 94.1 |
| *ND5* | 6130–7803 | 1674 | − | 5 | ATT/TAA | 84.3 |
| *trnHis (H)* | 7822–7886 | 65 | − | 18 | GTG | 84.6 |
| *ND4* | 7889–9199 | 1311 | − | 2 | ATA/TAA | 83.5 |
| *ND4L* | 9220–9507 | 288 | − | 20 | ATT/TAA | 86.8 |
| *trnThr (T)* | 9510–9575 | 66 | + | 2 | TGT | 90.9 |
| *trnPro (P)* | 9575–9641 | 67 | − | −1 | TGG | 83.6 |
| *ND6* | 9645–10212 | 568 | + | 3 | ATG/T − | 86.6 |
| *CYTB* | 10210–11349 | 1140 | + | −3 | ATT/TAA | 77.3 |
| *trnSer2 (UCN)* | 11348–11412 | 65 | + | −2 | TGA | 81.5 |
| *ND1* | 11395–12333 | 939 | − | −18 | ATA/TAA | 80.2 |
| *trnLeu1 (CUN)* | 12334–12402 | 69 | − | 0 | TAG | 87.0 |
| *rrnL* | 12403–13798 | 1396 | − | 0 | − | 86.9 |
| *trnAla (A)* | 13799–13873 | 75 | − | 0 | TGC | 84.0 |
| *trnGln (Q)* | 13879–13946 | 68 | + | 5 | TTG | 91.2 |
| *rrnS* | 13947–14802 | 856 | − | 0 | − | 88.4 |
| *trnMet (M)* | 14803–14871 | 69 | − | 0 | CAT | 84.1 |
| Control region | 14872–15549 | 678 | + | 0 | − | 84.4 |

**Table 4  Mitochondrial genome structure of *Ceroplastes rubens*.**

| Gene | Position (bp) | Size (bp) | Direction | Intergenic nucleotides | Anti- or start/ stop codons | A+T% |
|---|---|---|---|---|---|---|
| *COX1* | 1–1527 | 1527 | + | 42 | ATA/TAA | 80.4 |
| *trnLeu2 (UUR)* | 1532–1600 | 69 | + | 4 | TAA | 88.4 |
| *COX2* | 1601–2261 | 661 | + | 0 | ATA/T − | 83.4 |
| *trnLys (K)* | 2262–2328 | 67 | + | 0 | CTT | 83.6 |
| *trnAsp (D)* | 2325–2383 | 59 | + | −4 | GTC | 93.2 |
| *ATP6* | 2411–3091 | 681 | − | 27 | ATA/TAA | 89.7 |
| *COX3* | 3118–3891 | 774 | + | 26 | ATA/TAA | 86.3 |
| *trnGly (G)* | 3894–3950 | 57 | + | 2 | TCC | 94.7 |
| *ND3* | 3951–4286 | 336 | + | 0 | ATA/TAA | 90.8 |
| *trnAla (A)* | 4291–4350 | 60 | − | 4 | TGC | 91.7 |
| *trnAsn (N)* | 4370–4424 | 55 | + | 83 | GTT | 87.3 |
| *trnSer1 (AGN)* | 4424–4469 | 46 | + | −1 | GCT | 80.4 |
| *trnGlu (E)* | 4469–4522 | 54 | + | −1 | TTC | 94.4 |
| *trnTrp (W)* | 4527–4577 | 51 | + | 4 | TCA | 94.1 |
| *ND5* | 4579–6189 | 1611 | − | 56 | ATT/TAA | 88.3 |
| *trnHis (H)* | 6264–6320 | 57 | − | 74 | GTG | 89.5 |
| *ND4* | 6325–7605 | 1281 | − | 4 | ATA/TAA | 89.4 |
| *ND4L* | 7619–7963 | 345 | − | 13 | ATT/TAG | 92.2 |
| *ND6* | 7980–8375 | 396 | + | 16 | ATA/TAA | 89.6 |
| *trnPro (P)* | 8375–8433 | 59 | − | −1 | TGG | 89.8 |
| *ATP8* | 8435–8524 | 90 | − | 1 | ATA/TAA | 90.0 |
| *trnIle (I)* | 8546–8612 | 67 | + | 21 | GAT | 86.6 |
| *ND2* | 8613–9551 | 939 | + | 0 | ATT/TAA | 91.5 |
| *trnTyr (Y)* | 9558–9606 | 49 | − | 6 | GTA | 87.8 |
| *trnThr (T)* | 9608–9659 | 52 | + | 1 | TGT | 90.4 |
| *CYTB* | 9660–10736 | 1077 | + | 0 | ATC/TAA | 85.0 |
| *trnGln (Q)* | 10745–10796 | 52 | − | 8 | TTG | 92.3 |
| *ND1* | 10823–11728 | 906 | − | 86 | ATT/TAG | 86.5 |
| *rrnL* | 11729–12991 | 1263 | − | 0 | − | 90.7 |
| *rrnS* | 12992–13578 | 587 | − | 0 | − | 87.9 |
| Control region 1 | 13579–14408 | 830 | + | 0 | − | 85.4 |
| *trnPhe (F)* | 14409–14476 | 68 | − | 0 | GAA | 79.4 |
| Control region 2 | 14477–15276 | 800 | + | 0 | − | 88.4 |
| *trnMet (M)* | 15277–15345 | 69 | + | 0 | CAT | 82.6 |

rearrangement events are expected to occur very frequently in other unsequenced scale insects.

## Protein-coding genes

The 13 PCGs of *U. yanonensis* were similar in size to those of *P. citri*, without truncated or duplicated PCGs (Tables 2 and 3). However, most PCGs of *C. rubens* were shorter than *U. yanonensis* and *P. citri*, especially for *ATP8* and *ND6* (Fig. 6). Most PCGs of the

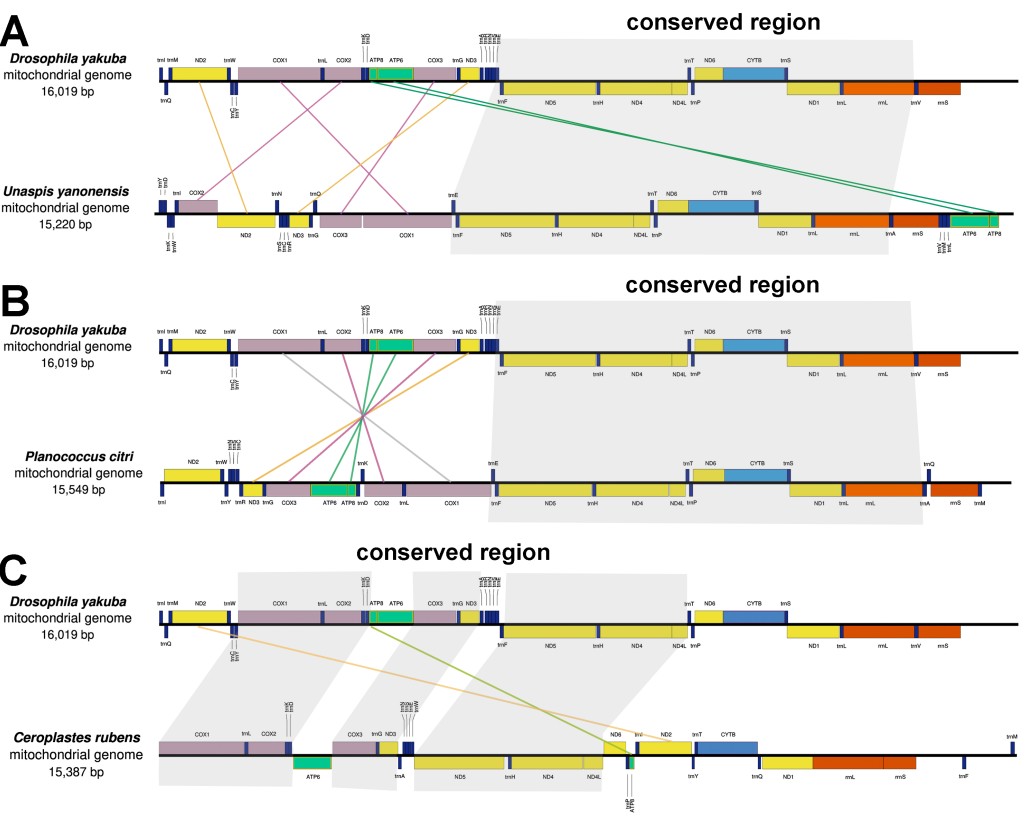

**Figure 2** Gene arrangements of *Unaspis yanonensis*, *Planococcus citri* and *Ceroplastes rubens* in comparison with *Drosophila yakuba*. (A) *Unaspis yanonensis*; (B) *Planococcus citri*; (C) *Ceroplastes rubens*. Conserved gene arrangements are covered in grey areas.

three mitogenomes utilized the standard ATN start codon (ATA, ATT, ATC and ATG). However, the special start codon TTG was used by *COX1* of *U. yanonensis* (Table 2). Twelve PCGs of each mitogenome had the complete termination codon TAN (TAA or TAG), whereas *ND6* of *U. yanonensis* and *P. citri* and *COX2* of *C. rubens* ended with an incomplete stop codon T. In the previously sequenced scale insect, *C. japonicus*, *COX2* also ended with an incomplete T (*Deng, Lu & Huang, 2019*). The relative synonymous codon usage (RSCU) values were calculated for the three mitogenomes (Fig. 7). In *U. yanonensis*, the most frequently used codon was TTA (Leu) whereas CTG(Leu), TCC(Ser), ACC(Thr), ACG(Thr), GCC(Ala), CAG(Gln), TGC(Cys), CGG(Arg) and AGC(Ser) were not used. In *P. citri*, the mostly used codon was also TTA (Leu), but CTC (Leu), AGC (Ser) and CGC (Arg) were the least. In *C. rubens*, TTA (Leu) was the most frequently used codon.

To evaluate the evolutionary rates of the PCGS, the average ratio of Ka/Ks was calculated for each PCG of the three mitogenomes (Fig. 8). The results showed that *ND4L* had the highest evolutionary rate, followed by *ATP8* and *ND5*, while *COX1* and *CYTB* appeared to be the lowest. The ratios of Ka/Ks were above 1 for most PCGs except for *COX1* and *CYTB*, suggesting that these genes are evolving under positive selection. However, the ratios of Ka/Ks for *COX1* and *CYTB* were below 1, indicating the purifying selection in these genes.

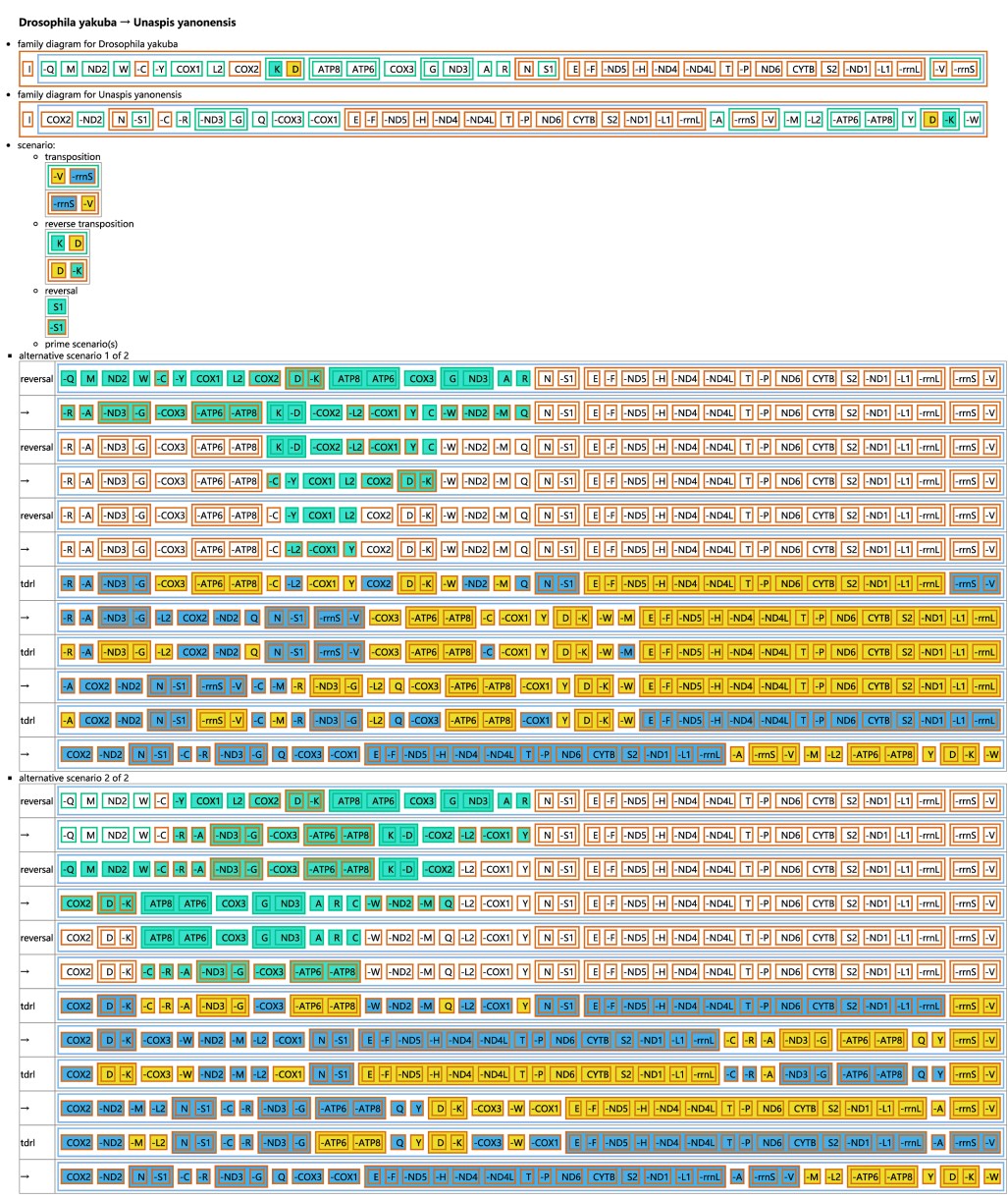

**Figure 3  Reconstruction of mitochondrial gene rearrangement scenarios in the evolution of *Unaspis yanonensis*.** The tRNA genes are represented by the amino acid abbreviations.

The two genes, *COX1* and *CYTB* which with relatively slow evolutionary rates have already been used as efficient phylogenetic markers in insects.

## Transfer RNA genes

The typical set of 22 tRNA genes were all detected in the mitogenome of *U. yanonensis*, but *trnV* was absent from the mitogenome of *P. citri* (Figs. 9 and 10). In *C. rubens*, only 17 tRNA genes were recognized and the three tRNA genes *trnA*, *trnQ* and *trnW* were manually predicted (Fig. 11). Length and A+T content of the tRNA genes were subequal between

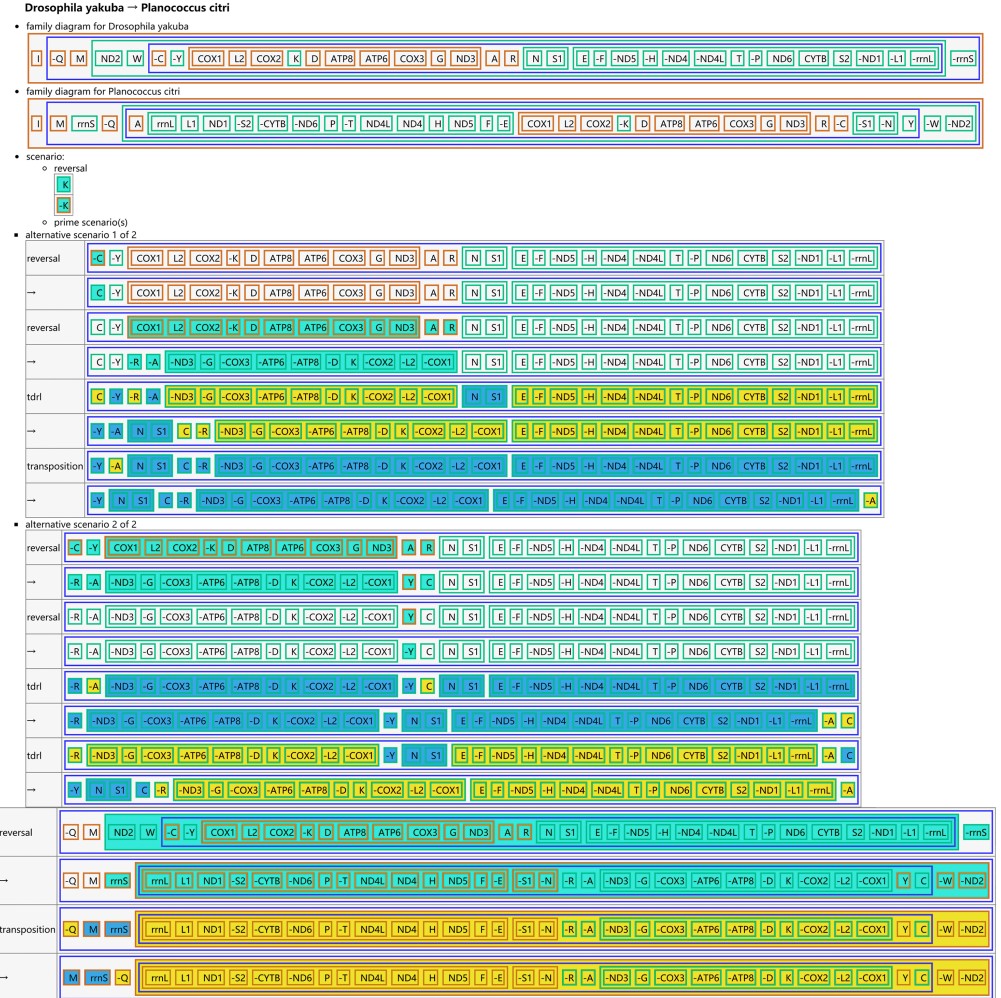

**Figure 4** **Reconstruction of mitochondrial gene rearrangement scenarios in the evolution of _Planococcus citri_.** The tRNA genes are represented by the amino acid abbreviations.

_U. yanonensis_ and _P. citri_, whereas the lengths of tRNA genes of _C. rubens_ were generally shorter than _U. yanonensis_ and _P. citri_. Individual tRNA gene of the three mitogenomes ranged in size from 49 to 75 bp; the longest tRNA gene was _trnA_ in _P. citri_ (Table 3); the shortest tRNA gene was _trnY_ in _C. rubens_ (Table 4). In the mitogenomes of _U. yanonensis_ and _P. citri_, most of the tRNA genes could fold into cloverleaf secondary structures, but the dihydrouridine (DHU) arms of _trnR_ and _trnS1_ were consistently lost. In _C. rubens_, most tRNA genes exhibited reduced DHU arms or T ψC arms. Such reductions of DHU arms were also reported in the tRNA genes of _S. coffeae_ (_Lu, Huang & Deng, 2020_), suggesting that tRNA gene reduction could be a very common phenomenon in the mitogenomes of scale insects. The anticodons of the tRNA genes were identical among the three scale insects. In the tRNA genes of _U. yanonensis_ and _P. citri_, a total of 12 and 19 mismatched
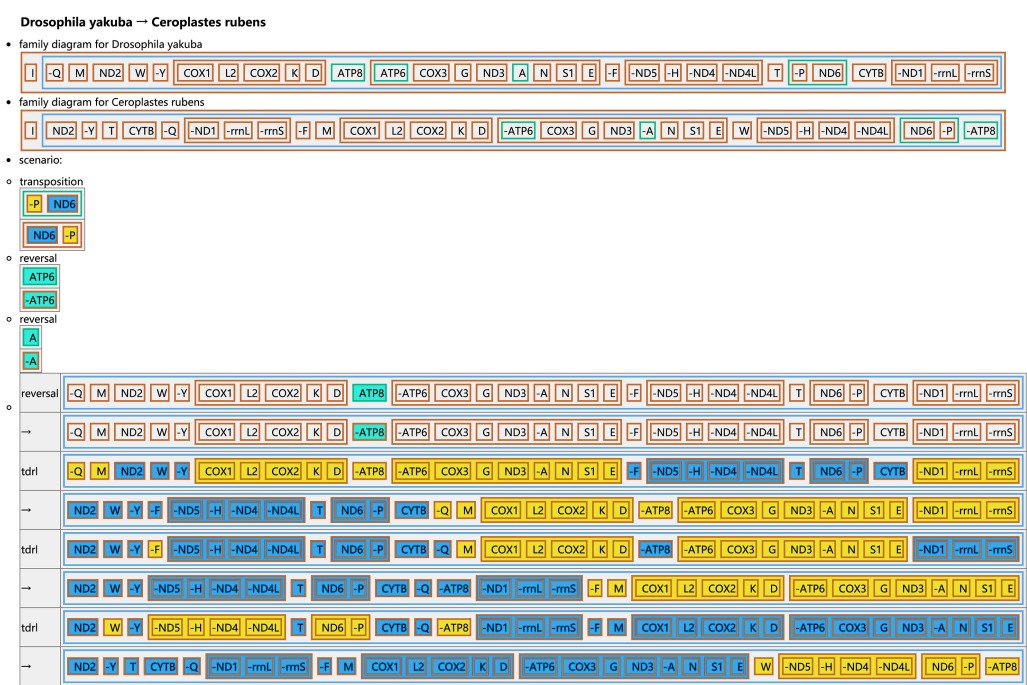

**Figure 5** Reconstruction of mitochondrial gene rearrangement scenarios in the evolution of *Ceroplastes rubens*. The tRNA genes are represented by the amino acid abbreviations.

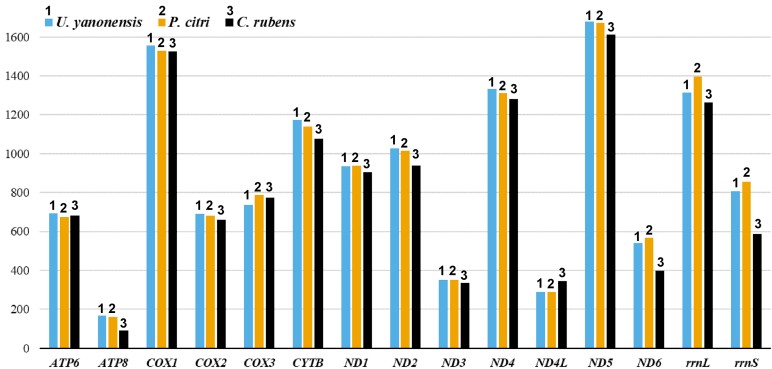

**Figure 6** Comparison of the length for each PCG and rRNA gene in *Unaspis yanonensis*, *Planococcus citri* and *Ceroplastes rubens*.

base pairs were respectively identified and all of them were G-U pairs. In *C. rubens*, only four mismatched G-U pairs were identified.

## Ribosomal RNA genes

There were two rRNA genes identified in in each mitogenome. The length and A+T content of each rRNA gene were subequal between *U. yanonensis* and *P. citri*, but the lengths of rRNA genes were much shorter in *C. rubens* (Tables 2–4). In *U. yanonensis*, the large ribosomal RNA (*rrnL*) gene was 1314 bp with an A+T content of 89.6%; the
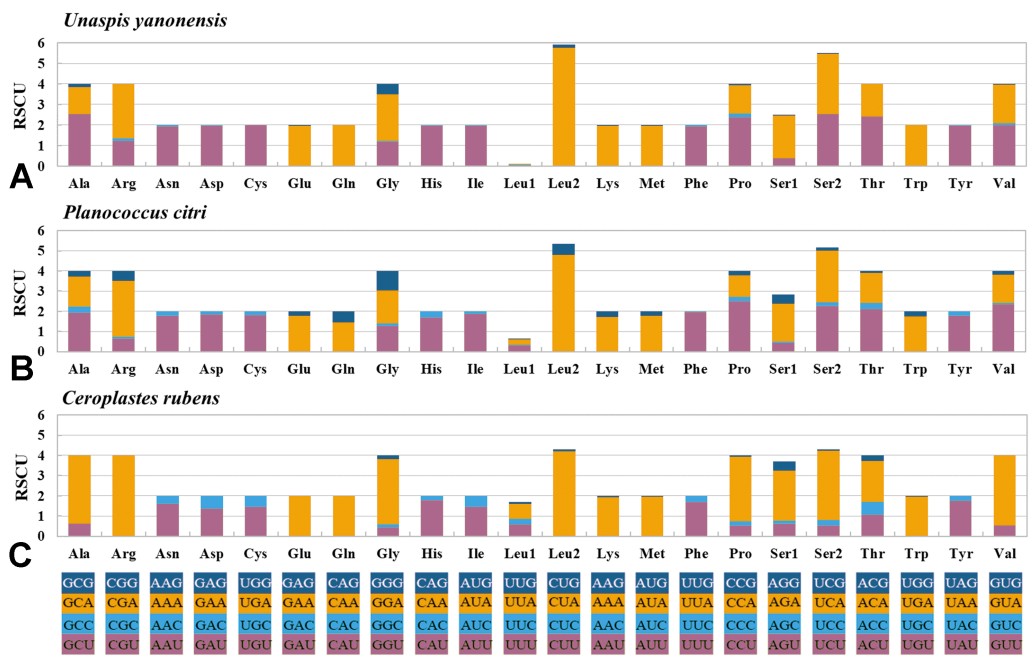

**Figure 7 Relative synonymous codon usage (RSCU) of PCGs in *Unaspis yanonensis*, *Planococcus citri* and *Ceroplastes rubens*.** (A) *Unaspis yanonensis*; (B) *Planococcus citri*; C: *Ceroplastes rubens*. Full codon families are indicated below the *X*-axis.

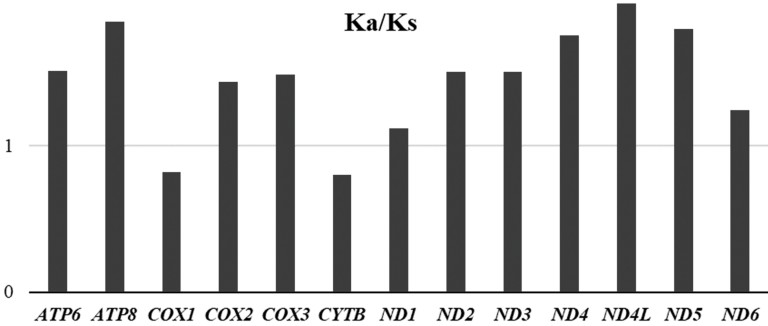

**Figure 8 Average evolutionary rates of PCGs in *Unaspis yanonensis*, *Planococcus citri* and *Ceroplastes rubens*.** The bar indicates each gene's Ka/Ks value.

small ribosomal RNA (*rrnS*) gene was 807 bp with a high A+T content of 90.6%. In *P. citri*, the *rrnL* gene was 1396 bp with an A+T content of 86.9%; the *rrnS* gene was 856 bp with an A+T content of 88.4%. In *C. rubens*, the *rrnL* gene was 1,263 bp with a high A+T content of 90.7%; the *rrnS* gene was 587 bp with an A+T content of 87.9%. Locations of the two rRNA genes were similar to *D. yakuba*, being neighbored with the *CYTB-ND1* PCG cluster (Fig. 2). Instead of the commonly found *trnV* between the *rrnL* and *rrnS* genes in other insects, the intermediate tRNA gene between the two rRNA genes was *trnA* in *U. yanonensis*, *trnA* and *trnQ* in *P. citri*, and completely absent in *C. rubens*.

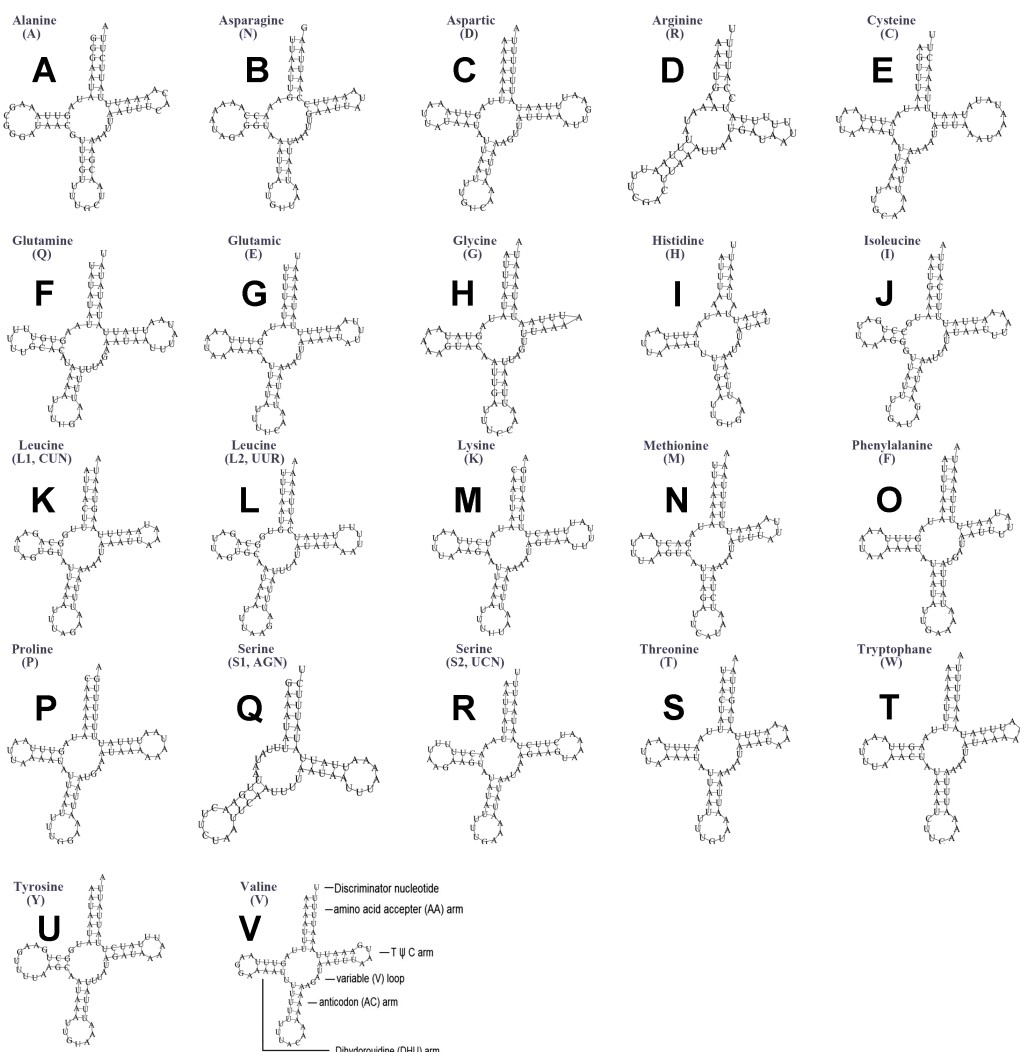

**Figure 9 Secondary structures of tRNA genes in the mitogenome of *Unaspis yanonensis*.** (A) trnA (Alanine); (B) trnN (Asparagine); (C) trnD (Aspartic acid); (D) trnR (Arginine); (E) trnC (Cystine); (F) trnQ (Glutamine); (G) trnE (Glutamic acid); (H) trnG (Glycine); (I) trnH (Histidine); (J) trnI (Isoleucine); (K) trnL1(CUN) (Leucine); (L) trnL2(UUR) (Leucine); (M) trnK (Lysine); (N) trnM (Methionine); (O) trnF (Phenylalanine); (P) trnP (Proline); (Q) trnS1(AGN) (Serine); (R) trnS2(UCN) (Serine); (S) trnT (Threonine); (T) trnW (Tryptophan); (U) trnY (Tyrosine); (V) trnV (Valine). The tRNA genes are labelled with their corresponding amino acids.

## Control region

Control region (CR), also known as A+T rich region, was the longest and most variable non-coding area in the three mitogenomes (Fig. 12). The CR of *U. yanonensis* was short with only 260 bp, being located between *trnY* and *ATP8* and with a relatively high A+T content of 81.9% (Table 2). The CR of *P. citri* was much longer than *U. yanonensis* (678 bp), being located between *trnM* and *trnI* and with an A+T content of 84.4% (Table 3). Two putative CRs were found in the mitogenome of *C. rubens*: the 830-bp long CR1 between *rrnS* and *trnF* and the 800-bp long CR2 between *trnF* and *trnM* (Table 4). A+T content

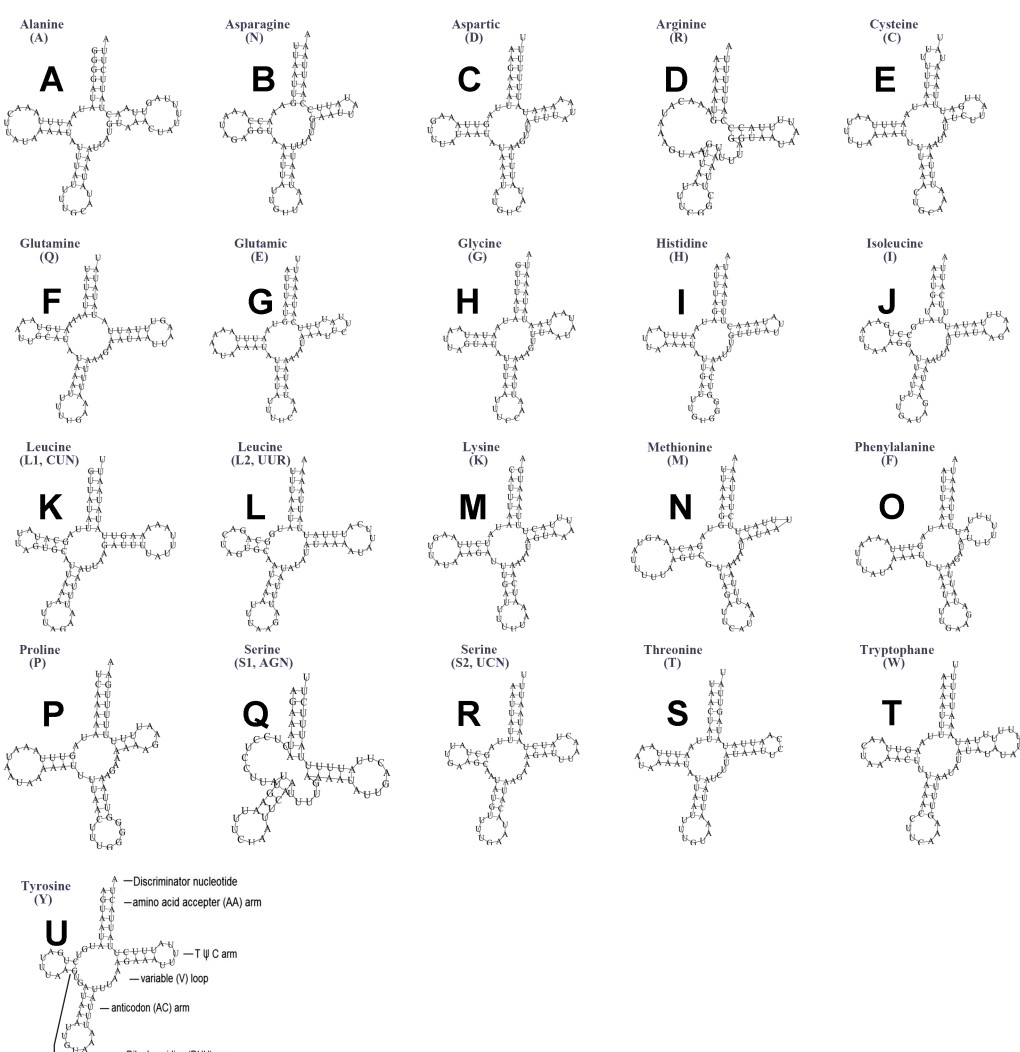

**Figure 10  Secondary structures of tRNA genes in the mitogenome of *Planococcus citri*.** (A) trnA (Alanine); (B) trnN (Asparagine); (C) trnD (Aspartic acid); (D) trnR (Arginine); (E) trnC (Cystine); (F) trnQ (Glutamine); (G) trnE (Glutamic acid); (H) trnG (Glycine); (I) trnH (Histidine); (J) trnI (Isoleucine); (K) trnL1(CUN) (Leucine); (L) trnL2(UUR) (Leucine); (M) trnK (Lysine); (N) trnM (Methionine); (O) trnF (Phenylalanine); (P) trnP (Proline); (Q) trnS1(AGN) (Serine); (R) trnS2(UCN) (Serine); (S) trnT (Threonine); (T) trnW (Tryptophan); (U) trnY (Tyrosine). The tRNA genes are labelled with their corresponding amino acids.

of the two CRs was 85.4% and 88.4%, respectively, higher than *U. yanonensis* and *P. citri*. The CR of *C. japonicus* and *S. coffeae* was 507 bp and 1454 bp, respectively, indicating the highly variable length of CRs in scale insects (*Deng, Lu & Huang, 2019*; *Lu, Huang & Deng, 2020*).

The CR of *U. yanonensis* was composed of 2.9 copies of tandem repeats; the first two copies had a consensus size of 91 bp, whereas the third repeat was 78 bp in length. The CR of *P. citri* contained three types of secondary structures that might function in regulating the replication and transcription of the mitogenome, including 2.3 copies of 110-bp long

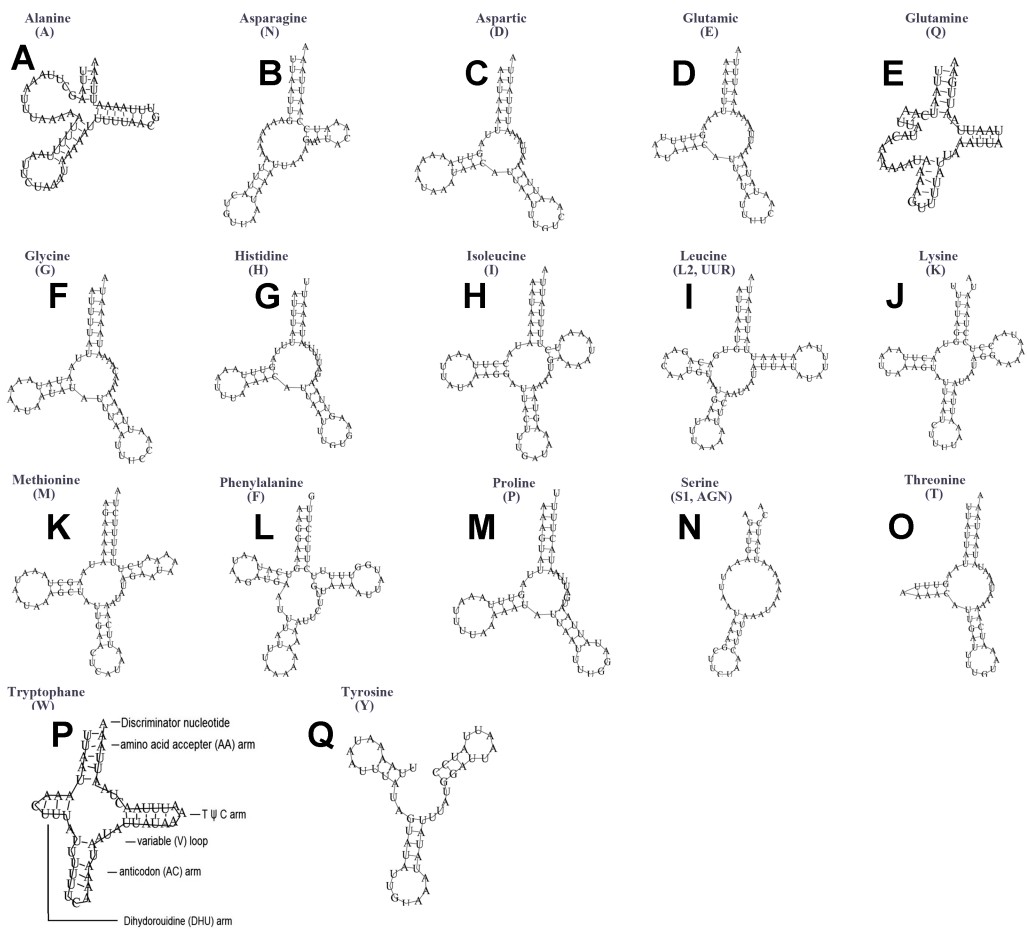

**Figure 11** **Secondary structures of tRNA genes in the mitogenome of *Ceroplastes rubens*.** (A) trnA (Alanine); (B) trnN (Asparagine); (C) trnD (Aspartic acid); (D) trnE (Glutamic acid); (E) trnQ (Glutamine); (F) trnG (Glycine); (G) trnH (Histidine); (H) trnI (Isoleucine); (I) trnL2(UUR) (Leucine); (J) trnK (Lysine); (K) trnM (Methionine); (L) trnF (Phenylalanine); (M) trnP (Proline); (N) trnS1(AGN) (Serine); (O) trnT (Threonine); (P) trnW (Tryptophan); (Q) trnY (Tyrosine). The tRNA genes are labelled with their corresponding amino acids.

tandem repeats, one 40-bp long poly-[TA]n stretch, and a 21-bp long stem-loop (SL) structure. The SL structure was initiated by a "TAA" motif and ended with a "GTA" motif. The longer tandem repeats and extra secondary structures of *P. citri* resulted in the longer CR than that of *U. yanonensis*. The CR1 of *C. rubens* contained 3.6 copies of 33-bp long tandem repeats but had no SL structures. The CR2 of *C. rubens* included 5 copies of 24-bp long tandem repeats and a combined SL structure. The length, nucleotide composition, number and types of structural elements in CRs of the three mitogenomes were found highly variable, which implied that the scale insect mitogenomes were likely to be regulated in different ways during the mitogenomic replication and transcription processes.

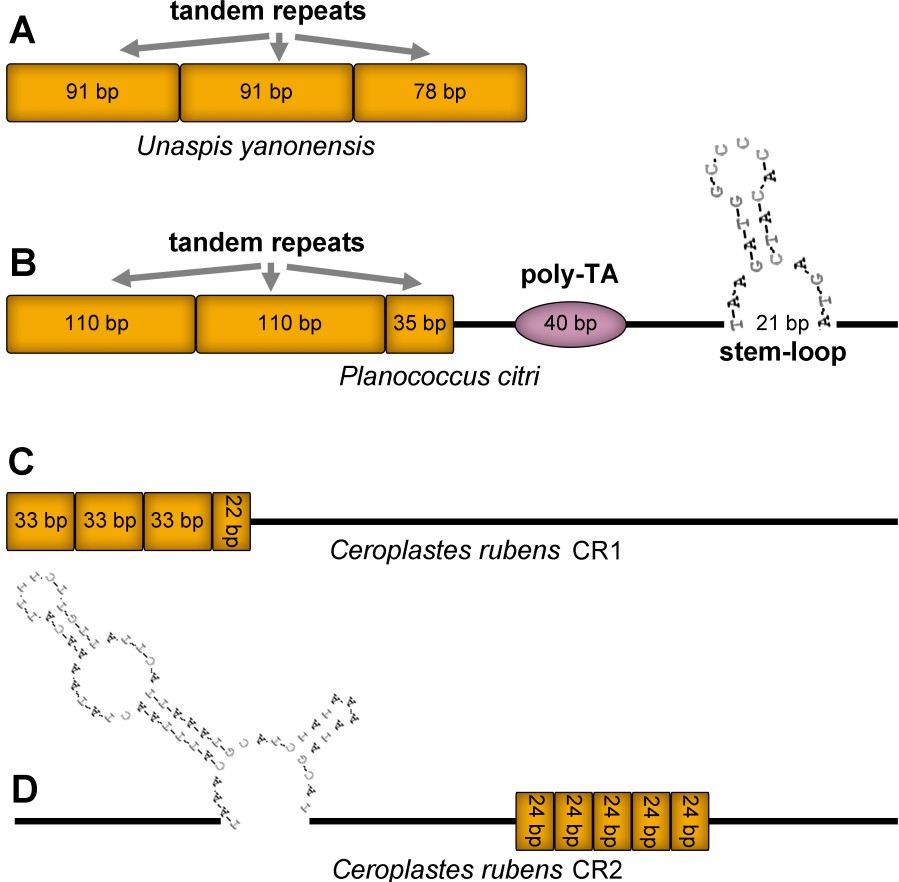

**Figure 12 Predicted structural elements in the control regions of *Unaspis yanonensis*, *Planococcus citri* and *Ceroplastes rubens*.** Tandem repeat units are indicated by orange boxes. Poly-[TA]n stretch is indicated with purple ellipse. Stem-loop structure is indicated by its shape and base pairs.

## DISCUSSION

To test the reliability of the three sequenced mitogenomes and investigate the mitochondrial phylogenetic relationships within Coccoidea, nucleotide sequences of available scale insects were obtained from GenBank and used in the phylogenetic analyses (Table 1). The two phylogenetic trees using BI and ML analyses generated identical topological structures for Coccoidea (Fig. 13). The three families of Coccoidea were grouped together, suggesting the probable monophyly of Coccoidea as found in *Von Dohlen & Moran (1995)*, which used the small-subunit (18S) ribosomal DNA in the phylogenetic analysis. The monophyly of Coccidae was supported with high values, indicating the efficiency of mitogenome data in grouping members of the same family and partially supporting the correctness of the tree topologies. Pseudococcidae was recovered as the sister group of Diaspididae and the phylogenetic position of their combined clade was supported basal to Coccidae. However, in previous molecular and morphological studies (*Gullan & Cook, 2007*; *Cook, Gullan & Trueman, 2002*; *Hodgson & Hardy, 2013*), Pseudococcidae was supported basal to Coccidae

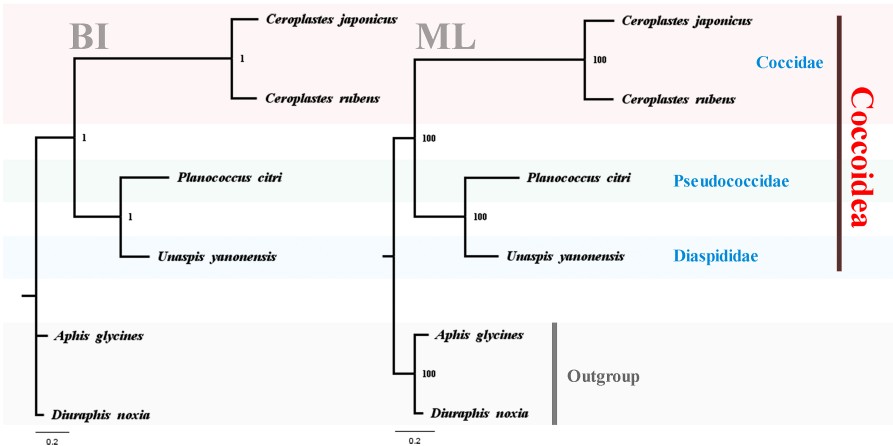

**Figure 13 Phylogenetic relationships within Coccoidea inferred by Bayesian inference and maximum likelihood analysis.** Numbers at the nodes are posterior probabilities and bootstrap values. The family names are listed after the species.

and Diaspididae. The insufficient mitogenome data of Coccoidea, and the selection of different taxa and different molecular markers in the phylogenetic analysis were very likely to cause different phylogenetic results especially for the family levels (*Chen et al., 2018*). The new mitogenome data obtained in this study provided a basis for the accurate reconstruction of mitochondrial phylogeny in Coccoidea. The sequencing of more scale insects in future can also provide new data for our understanding of the highly rearranged mitogenomes and evolutionary history of these enigmatic insects. Sufficient representatives and molecular data will furtherer resolve the inner relationship of Coccoidea.

## CONCLUSIONS

The complete mitochondrial genomes of *U. yanonensis*, *P. citri* and *C. rubens* were sequenced and analyzed. The mitochondrial genes of the three scale insects were highly rearranged and different from other scale insects. The phylogenetic reconstructions with BI and ML methods generated identical phylogenetic topology and supported the inner relationship of Coccoidea as Coccidae + (Pseudococcidae + Diaspididae). More mitogenomes should be obtained in future works to resolve the phylogeny of scale insects.

## ACKNOWLEDGEMENTS

The authors thank the editor (Mikhail Gelfand) and reviewers (Nina Voronova and another anonymous reviewer) for valuable comments and improvement of the manuscript.

### Funding

This research was supported by the National Key R & D Program of China (No. 2017YFD0202002), Research on green pest control technology of characteristic fruit

(No. 2016GYSH-018), and Sichuan fruit innovation team of national modern agricultural industry technology system (No. 2019-2023). The funders had no role in study design, data collection and analysis, decision to publish, or preparation of the manuscript.

### Grant Disclosures

The following grant information was disclosed by the authors:
National Key R & D Program of China: 2017YFD0202002.
Research on green pest control technology of characteristic fruit: 2016GYSH-018.
Sichuan fruit innovation team of national modern agricultural industry technology system: 2019-2023.

### Competing Interests

The authors declare there are no competing interests.

### Author Contributions

- Hong-Ling Liu conceived and designed the experiments, analyzed the data, prepared figures and/or tables, and approved the final draft.
- Qing-Dong Chen performed the experiments, prepared figures and/or tables, and approved the final draft.
- Song Chen conceived and designed the experiments, performed the experiments, prepared figures and/or tables, and approved the final draft.
- De-Qiang Pu and Zhi-Teng Chen analyzed the data, authored or reviewed drafts of the paper, and approved the final draft.
- Yue-Yue Liu and Xu Liu conceived and designed the experiments, authored or reviewed drafts of the paper, and approved the final draft.

### DNA Deposition

The following information was supplied regarding the deposition of DNA sequences:
The mitogenome sequences of *U. yanonensis* (MT611525), *P. citri* (MT611526) and *C. rubens* (MT677923) are available at GenBank.

### Data Availability

The complete sequences are available in the Supplemental Files.

### Supplemental Information

Supplemental information for this article can be found online at http://dx.doi.org/10.7717/peerj.9932#supplemental-information.

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
