# Peer review of "The highly rearranged mitochondrial genomes of three economically important scale insects and the mitochondrial phylogeny of Coccoidea (Hemiptera: Sternorrhyncha)"

_PeerJ, doi:10.7717/peerj.9932_

## Round 0.1 · original submission · Major Revisions

Most reviewer's comments are editorial; however, there is a substantial criticism of the phylogenetic analysis that needs to be addressed in revision.

·

Basic reporting

No comment

Experimental design

No comment

Validity of the findings

No comment

Additional comments

Line 42. Please, rephrase “Coccoidea exhibit complete metamorphosis” as so it can mislead readers in relation to the Coccoidae belonging to Hemi- or Holometabola.
Line 159-160. Please, rephrase “The CREx analysis predicted the alternative scenarios how the three mitogenomes rearranged from that of D. yakuba”, otherwise it can be understood that D. yakuba is an ancestral organism for Coocoidae as the authors only speak about the ancestral type of mitochondrial genome.
Line 192-194. Genes COI as well as cytB have been used as phylogenetic markers for insects for a long time and their efficiency is already confirmed and widely recognized. I would recommend removing this phrase or, please, specify the underlying ideas. This remark remains at the discretion of the authors.
Line 200-201. Please, add “C. rubens” into the sentence: “Length and A+T content of the tRNA genes were subequal between U. yanonensis and P. citri, whereas the lengths of tRNA genes were generally shorter than U. yanonensis and P. citri.” In this sentence it is not obvious what species’ tRNA genes are shorter than U. yanonensis and P. citri.
Line 247-250. “The length, nucleotide composition, number and types of structural elements in CRs of the three mitogenomes 249 were found highly variable, which implied that the scale insect mitogenomes were regulated in different ways during the mitogenomic replication and transcription processes”. I am not sure that the difference in nucleotide organization of CR allows making conclusions about the difference in the replication/transcription regulation even though this output is logical. This remark remains at the discretion of the authors.

Reviewer 2 ·

Basic reporting

No comment

Experimental design

No comment

Validity of the findings

No comment

Additional comments

Please find herewith a few observations;
The scale insects are hard to deal with because of its waxy surface coating; especially for molecular work, as it can act as a PCR inhibitor. Overall, this paper is very informative except for the phylogenetic analyses.
To me, it is highly immature to state the monophyly of Aphidoidea by merely analyzing a few aphid species mitogenome (that too Aphididae alone and has nothing to do with the paper!!!). The same applies to the other superfamilies as well, as all of them were underrepresented with very few species. Hard to understand why the authors employed the whole of Sternorrhyncha, for their phylogeny? Is it merely to increase the sample size? (The whole paper is on Scale insect, except this analysis!)
It is better to do the phylogenetic analyses using the coccids from this work and publicly available along with the aphids (a couple of species) as outgroups rather than the present one.
Otherwise, the paper looks solid with mitogenome analyses, figures (except phylogeny), and tables.
Few minor corrections have been added in the PDF version (please find the attachment).
Best regards,

Annotated reviews are not available for download in order to protect the identity of reviewers who chose to remain anonymous.

---

## Round 0.2 · accepted · Accept

The reviewers are satisfied with the revised version. However, I'd suggest proofreading the manuscript to improve English (one random example of an incorrect sentence: "Despite of the multiple tRNA gene rearrangements")

·

Basic reporting

no comment

Experimental design

no comment

Validity of the findings

no comment

Additional comments

This manuscript can be accepted as it is.